# Implications of Cash Transfer Programs for Mental Health Promotion among Families Facing Significant Stressors: Using Ecological Systems Theory to Explain Successes of Conditional and Unconditional Programs

**DOI:** 10.3390/bs14090770

**Published:** 2024-09-02

**Authors:** Tali L. Lesser, Maya Matalon, Caroline S. Clauss-Ehlers

**Affiliations:** Department of Psychology, School of Health Professions, Long Island University Brooklyn, Brooklyn, NY 11201, USA

**Keywords:** cash transfer programs, COVID-19, family resilience, mental health, socioecological systems theory

## Abstract

The purpose of this paper is to apply Bronfenbrenner’s ecological systems theory to explore the literature on how Conditional Cash Transfer (CCT) and Unconditional Cash Transfer (UCT) programs might support positive mental health outcomes. The paper begins with transnational considerations of stress, such as poverty and COVID-19, and their impact on mental health. Bronfenbrenner’s theory is applied to better understand the mechanisms by which CCT and UCT programs can potentially lead to positive outcomes for children and families who face such stressors. The implications of cash transfer programs are subsequently discussed in terms of how they might promote positive mental health outcomes among families globally. This theoretical application paper concludes with a call for transnational research to explore connections between cash transfer programs and mental health outcomes for children/adolescents and their parents/caregivers.

## 1. Introduction

To apply ecological systems theory to explore how Conditional Cash Transfer (CCT) and Unconditional Cash Transfer (UCT) programs might support positive mental health outcomes, we first begin with global considerations of stressors such as poverty and COVID-19 and their impact on mental health. The paper will then consider the possible mechanisms by which these cash transfer programs can impact caregiver and child mental well-being, as well as family resilience, through a theoretical application of Bronfenbrenner’s ecological model.

*Poverty.* Children globally are disproportionately affected by extreme poverty. In 2022, it was estimated that 333 million children lived on less than $2.15 per day, which is considered the extreme international poverty line by the World Bank [1]. Children are more than twice as likely as adults to live in extreme poverty, although their share of the population is only 31%. Between 2013 and 2023, there was an estimated reduction in the extreme child poverty rate from 20.7% to 15.9% [1]. However, it is estimated that COVID-19-related disruptions severely impeded progress in reductions in child poverty and that 30 million more children were projected to be lifted out of extreme poverty had COVID-19 not occurred [1].

It has been well documented in the literature that childhood poverty may be linked to poorer mental health outcomes in children and adolescents [2,3,4,5,6]. For instance, studies conducted in North America, Europe, and Australia demonstrated that children and adolescents with indicators of low socioeconomic status (SES) are two to three times more likely to develop mental health problems [2,3]. A retrospective study in a US national sample found that childhood financial hardship, which leads to having fewer material resources, was strongly associated with the onset of mental health problems but had no impact on their course or severity [4]. Meanwhile, low parental education, which was unrelated to the onset of disorders, was significantly associated with disorder persistence and severity. Results suggest that higher parental education may indicate greater access to resources such as mental health treatment and reduced stigma toward mental illness [4,5]. In a longitudinal study of families residing in rural areas in the Northeastern United States, early childhood poverty from ages 9 to 24 significantly predicted greater cumulative risk exposure over time, which in turn significantly predicted greater internalizing (e.g., depression and anxiety) and externalizing symptoms (e.g., aggression and conflict) [6]. Another longitudinal study using a nationally representative sample of youth in the U.S. found that children in relative poverty ages 5 to 12 had more socioemotional problems, such as anxiety, depression, antisocial behavior, peer conflict, and social withdrawal, than children not in relative poverty. Additionally, children had lower social–emotional problems when mothers had more years of education, had higher cognitive test scores, and provided more emotional support [7].

*COVID-19.* As noted, the COVID-19 pandemic further exacerbated both financial and emotional stressors for children and their families. With the onset of the COVID-19 pandemic, many children and adolescents’ social contact was highly restricted to those within their families. Additionally, the global economic crisis caused by pandemic-related shutdowns and restrictions led to significant increases in unemployment and financial pressures on families [8,9]. A meta-analysis of cross-sectional community-based studies found that during the first six months of the pandemic, the prevalence of anxiety in the general population was 25%, which is approximately three times higher than the pre-pandemic global prevalence of anxiety disorders. Among the 43 studies across global contexts, anxiety appeared higher for those who suffered loss of income due to COVID-19, were unemployed, or perceived a high risk of losing their job [10].

The negative effects of child poverty on children’s health and well-being have been observed long prior to the onset of the COVID-19 pandemic. However, the COVID-19 pandemic exacerbated problems related to poverty and mental health. Children’s school and family lives were completely upended through school and community closures, which may have disrupted their relational health [11]. The closure of schools and other in-person activities may have led to feelings of loneliness and isolation, while some children may have compensated for these feelings through increased internet use and time spent with family at home [12]. However, social isolation and more time spent with family members may also lead to an increased risk of child maltreatment and domestic violence, leading to child mental health problems [9]. Childhood poverty and COVID-19 are two transnational considerations of stress that may have an impact on mental health and well-being in families, thus further supporting the case for structural programs to address them.

*Family stress model.* The family stress model [13] can be used to understand how the effects of poverty, COVID-19, and negative economic pressures lead to disruptions in family functioning and the quality of parent–child interactions [14]. This model describes how caregivers overwhelmed by the demands of poverty may be unable to meet the emotional and cognitive needs of their children. Additionally, poor mental health and a lack of access to treatment and/or parenting support may understandably interfere with the quality of parent–child interactions [13,15]. This becomes particularly relevant when considering that the caregiver–child relationship is an essential component of resilience. According to the Family Investment Model [16], parents who have access to greater resources can invest in their children through educational materials, such as reading books to them or taking them to museums, therefore protecting their children from the effects of poverty. In support of this model, one study found that parents’ educational expectations and specific parenting behaviors, such as play and reading, indirectly link the relationship between parents’ educational attainment and children’s academic achievement [17].

On a macrosystem level, structural strategies can function to address issues stemming from the negative effects of childhood poverty on children’s mental health and well-being. Cash transfer programs serve as governmental programs that are implemented widely across the globe. Though the aim of cash transfer programs is often on alleviating poverty, structural change in the form of these programs can also positively affect the mental health and well-being of children and families. There is an increased call for mental health impacts to be considered as part of evaluations of cash transfer programs [18], further supporting the view of cash transfer programs as a means for structural change.

Given transnational considerations of family stress such as poverty and COVID-19, the next section of the paper examines unconditional and conditional cash transfer programs as structural strategies to promote resilience. A global perspective is taken to explore how such programs are implemented in Latin America, Africa, India, and North America despite social, political, cultural, and economic differences in these regions.

## 2. Implications of Cash Transfer Programs for Mental Health

In the following sections, the authors discuss conditional and unconditional cash transfer programs implemented in Latin America, Africa, India, and North America. This discussion considers structural strategies as examples of policy mechanisms that seek to promote resilience and positive mental health outcomes in children and families. As explained further below, conditional cash transfer programs (CCTs) function by providing economic incentives through cash transfers to recipients in exchange for completing specific requirements or targeted behaviors [19]. In contrast, unconditional cash transfer programs (UCTs) provide economic incentives without any additional behavioral targets [19].

### 2.1. Conditional Cash Transfer Programs

Conditional Cash Transfer (CCT) programs are widely implemented globally, many in Low- and Middle-Income Countries (LMICs), with many programs existing in Latin America [20]. With CCTs, receipt of cash transfers is dependent upon the completion of a specific requirement (e.g., school attendance, regular health visits, etc.) [19]. As such, CCT programs offer incentives in exchange for parental investment in support of their child by including conditionalities, thus seeking to engage parents in structural systems such as education and healthcare in support of their children and, in turn, their families. Additionally, CCTs may influence family norms around education and healthcare utilization, thus enhancing routines and interactions with systems that have a direct impact on mental health and well-being [21].

Among two of the most well-studied and largest CCTs to date are government-sponsored programs in Latin America: Brazil’s *Bolsa Familia* and Mexico’s *Oportunidades* programs. Each program aims to end intergenerational cycles of poverty by enhancing human capital and increasing parental investment in their children’s education and health [22,23]. The programs provide a monthly sum on the condition of completing requirements in health, nutrition, and education [24,25,26]. Such requirements might include meeting targets in school attendance and showing receipts for recommended preventive healthcare visits and vaccinations [25,26].

Research has focused less directly on evaluating the impact these programs have on child and adult mental health. Additionally, there has been scant research on the pathways and mechanisms by which CCTs have had an impact on families [21]. Studies looking specifically at the impact of CCT programs on child and caregiver mental health and well-being have mixed results. For instance, children ages 4 to 6 receiving between 3.5 to 5 years of support from the *Oportunidades* program were found to have a 10% decrease in aggression and oppositional behaviors [27]. There was no association, however, between participation in *Oportunidades* and decreased symptoms of anxiety or depression, nor regarding overall behavioral problems [27]. Notably, cortisol levels, which are biological markers of increased stress, were significantly lower in a sample of 2- to 6-year-old children participating in *Oportunidades* [28].

While not much research has been conducted examining the impact of CCTs on caregiver mental health, one study found that women from poor rural areas of Mexico who participated in *Oportunidades* for 3.5 to 5 years had lower self-reported depressive symptoms [29]. Additionally, in another study, maternal depression acted as a moderator such that children of depressed mothers experienced the largest decreases in cortisol levels [28].

Other studies also demonstrate mixed findings regarding the impact of CCTs on child and adolescent mental health and well-being. Ziebold and colleagues (2021) examined the long-term impact of Bolsa Familia with a population-based sample of 11-year-olds in the urban area of Pelotas, Brazil, and they found that participation in the program at age 6 did not have a significant impact on early adolescent mental health [30]. Further, there was no impact on externalizing, prosocial and violent behaviors, and socioemotional competencies [30]. Similarly, research on adolescent mental health among recipients of a government-implemented CCT in Tanzania, Productive Social Safety Net (PSSN), found no overall intervention effects of the CCT on depressive symptoms among a full sample of youth aged 14 to 28 [31]. However, subsample results indicated that depressive symptoms among males reduced, while female depressive symptoms increased. The authors suggest that the negative impact of the CCT on females’ mental health was driven by the subsample of women who were ages 18 and older. These women might be more likely to spend an increased amount of time on domestic chores so that younger children in the family can attend school, which might have a negative impact on their mental health. Meanwhile, the men in the sample were positively impacted by the CCT, as they did not have the same societal expectations or domestic responsibilities as the women [31].

New York City’s Opportunity NYC-Family Rewards, a CCT program modeled after the *Oportunidades* program, was one of the first CCT programs to be evaluated in a randomized controlled study in the US [32]. However, when examining the possible program impact on mental health and indicators of mental well-being for parents, Riccio and colleagues (2010) found no significant effects. In addition, no programmatic impact was found concerning childhood ADHD or adolescent depression and anxiety [32]. Interestingly, in another study of the Opportunity NYC-Family Rewards Program, there were significant reductions in adolescent aggressive behaviors and substance use, though these were not direct targets of the program [33]. This finding is notable given that it reflects a similar trend of decreased aggressive behaviors in the *Oportunidades* program, albeit the *Oportunidades* program had a younger sample [27]. Additionally, the Opportunity NYC-Family Rewards Program reduced time spent with peers and increased time spent with family members, which may have implications for decreasing risky behaviors in peer groups [33]. Increased time spent with family members may have tradeoffs depending on the family context. As previously stated, more time spent with family may create higher vulnerability to exposure to abuse and domestic violence within certain families [9]. However, increased time spent with family members is a protective factor within the context of a positive family environment and is a positive implication of some CCTs [33].

### 2.2. Unconditional Cash Transfer Programs

In recent years, there has also been a broader focus on implementing Unconditional Cash Transfer (UCT) programs globally. Unlike CCTs, UCTs do not have any specific conditional requirements attached to the receipt of cash transfers. Current research looking at the effectiveness of adding a conditions requirement is inconclusive [20]. However, there are recent data from several studies across the globe, especially in low- or middle-income countries (LMICs), that have demonstrated that UCTs can have a positive effect on the mental well-being of children and families, among other positive health and long-term outcomes [34]. In contrast to research on CCTs, research on mental health impacts of UCTs has demonstrated promising results.

For instance, some studies have demonstrated a link between UCTs, as well as similar cash transfer programs, on improvements in components of mental well-being among parents [35,36]. Specifically, research on nationally implemented, government-sponsored UCTs for new parents in India [35] and for caregivers of young children living in economic poverty in South Africa [36] demonstrated that the UCTs had a significant impact on improving mental health through reductions in parental depression. Interestingly, the authors hypothesized that the UCT led to reductions in depression for new parents in India by alleviating debt resulting from medical expenses related to birth and delivery [35]. Similarly, the study examining the positive effects of a UCT program in South Africa on mental health was mediated by changes in physical health and income-related lifestyle changes [36]. A UCT in Ecuador, however, found no significant impact on maternal depression [37].

Other studies have also suggested that UCTs can enhance caregiver psychological well-being. A study looking at the effects of a UCT program in Zambia found that mothers’ self-reported happiness improved, and they had a more positive and hopeful outlook regarding their child’s well-being and future [38]. The authors suggest that this was due to how the women perceived their financial situation rather than changes in their absolute poverty [38]. Thus, the increased income may have had an impact on aspects of the women’s lives outside of their financial situation, such as increasing their positive outlook for the future, leading to enhanced mental health and markers of well-being.

A pilot study of a UCT for mothers facing economic challenges in New York City indicated that there may be evidence to suggest similar outcomes in urban areas of the United States [39]. Mothers were randomly assigned to receive $100 a month or $20 a month by cash transfer on debit cards. Results of the study indicated that the mothers reported the extra cash provided by the UCT transfer helped reduce the financial burdens during times of little or no income, even at the lower amount of $20. Additionally, the cash made a difference in being able to meet basic needs, with mothers mostly reporting being able to save it to use in significant moments of financial need such as transportation to the doctor’s office or spending it for items needed for their children. Notably, participants’ qualitative reports demonstrated that the cash transfers were vastly important in enhancing the participants’ psychological well-being and self-esteem as mothers [39].

Other studies that examined the benefits of UCTs for families have found a significant impact on the psychological well-being of children, as well as improved physical health and academic achievement [40]. For instance, one study examining two national government sponsored UCT programs delivered in Malawi for economically challenged households with children that had no labor opportunities or with children who had been orphaned found that the cash transfers were associated with improved mental health, including reduced odds of depression among adolescents [41]. Another study found similar outcomes for a UCT program in Kenya, demonstrating reductions of depression by 24% among adolescent recipients of the transfers; however, this was only significant for adolescent males [42]. The authors argue that the observed positive program impacts for adolescent males and not females may be explained by how adolescent females are more likely to be depressed [42]. Though this is not addressed by the authors, culture and societal expectations/norms may also have influenced why adolescent females in the study were more likely to be depressed and why the program did not significantly impact their mental health. For example, the expectation that adolescent women have increased responsibility in the home and care for younger siblings may be an obstacle to school enrollment and broader opportunities, which may impede mental health.

In the United States, Congress passed a temporary expansion of the Child Tax Credit (CTC) in March 2021 as part of the American Rescue Plan Act (ARP) [43]. The CTC expansion set out to help mitigate the negative effects of the pandemic on household finances. As a result, it in essence functioned as a UTC, since families received payments with no behavioral requirements. The CTC was originally enacted in 1997, and along with the Earned Income Tax Credit, has been more effective in decreasing the number of children living in poverty than any other economic subsidy efforts in the United States [43]. Notable revisions to the CTC as part of the ARP included increasing the amount of money families receive, monthly distribution, making credit fully refundable, and removing minimum income requirements. This increased accessibility to approximately 26 million children whose families were previously ineligible, especially Black and Latinx children [44].

Research has demonstrated mixed findings when examining the parent mental health benefits of the aforementioned CTC [45,46,47]. One study found fewer depressive and anxiety symptoms among parents in families with earnings below $35,000 as compared to families with higher earnings [46]. Parents of Black, Latinx, and other (non-Asian) racial and ethnic backgrounds demonstrated greater reductions in anxiety symptoms compared to non-Hispanic White parents [46]. Similarly, another study found evidence that the CTC reduced anxiety and depression symptoms in parents living in households facing economic challenges [47]. Additionally, the authors found that the CTC decreased anxiety symptoms after three months of payments, suggesting that it took time for mental health benefits to take effect. Other studies demonstrated null effects of the CTC on parental well-being [45,48]. Glasner and colleagues (2022) speculate that the temporary nature of the CTC expansion limited its impact because it may have influenced how recipients planned to use the money [45]. For example, uncertainty about the policy’s renewal may have increased recipients’ worries about meeting ongoing needs if benefits were to cease and may not have influenced recipients to make long-term investments (e.g., education, employment, etc.) that are likely to reduce symptoms of depression and anxiety. There is a gap in the research when examining the impact of the CTC on child mental health.

## 3. Applying Ecological Systems Theory to Child Development and Family Well-Being: Implications for Cash Transfer Programs

Given the current post-quarantine COVID-19 context that has indicated an increase in depression and anxiety symptoms among youth globally [8,49], an understanding of the protective factors associated with CCT and UCT programs presents critical implications. As mentioned, however, it is interesting that there is little empirical research investigating the mechanisms by which cash transfer programs have a positive impact on parent and child mental well-being [21]. This may be because many cash transfer programs have a target focus on poverty-related outcomes, such as education and physical health, rather than mental health. When considering the basic hierarchy of needs, responding to direct threats to individual and family well-being such as food insecurity, income, and physical health is necessary to address poverty; however, it is also important to recognize how mental well-being can shape and have an impact on long-term intergenerational cycles of poverty and thus consider how these structural solutions can impact these [50]. For instance, if a parent experiences depression to the point that it is difficult to function, the parent may lose their source of income if they are unable to get to work.

Figure 1 provides a visual application of Bronfenbrenner’s socioecological systems theory to various levels of child development and family well-being and engagement with the surrounding community. An application of the theory suggests that youth behavior and development are embedded among different spheres within a child’s life, namely the microsystem, exosystem, macrosystem, and chronosystem. Cash transfers intervene throughout multiple system levels, affecting the child, family, and community, as well as interactions between these. While it is evident how CCT and UCT programs function on a macrosystem level as structurally encompassing governmental programs, the socioecological model provides further understanding with regard to how cash transfer programs might have an impact on parent and child mental health and well-being across the different levels of the ecological framework.

### 3.1. Individual Level

On the individual level, both CCTs and UCTs can enhance important protective factors that might mitigate the impact of poverty on mental health such as increasing youth hopefulness, self-efficacy, self-esteem, and internal locus of control [21,22]. Several theoretical pathways have linked increased income to an increased sense of self-worth and self-efficacy. For instance, Adato et al. (2016) [51] cite social capital theory [52] as an important pathway in which these programs impact children’s resources and self-perceptions. In social capital theory, personal connections, such as membership in certain families, clubs, or groups, can lead to jobs and valuable connections, alleviating material constraints in an indirect way. Social capital theory can help us understand why some adolescents in extreme poverty may engage in risky sexual activity and unproductive peer groups, especially when these behaviors help offer an immediate way to alleviate material constraints or access self-worth [51].

While cash transfers tend to target basic needs, they can also enhance basic symbolic capital—resources that provide recognition of social status and prestige that are linked to culture and that provide children with a sense of self-worth and dignity when interacting with their peers [52]. Basic symbolic capital can include things like being able to bathe and show up to school clean. Heinrich and Hoddinott (2017) describe how poverty can lead to a sense of powerlessness and hopelessness regarding future economic outcomes and opportunities [53]. Cash transfers, even if merely covering some basic needs, may respond to perceptions of poverty by providing a sense of hope for the future and mitigating some of the negative effects that stress and hopelessness can have on psychosocial well-being.

For instance, research on adolescents whose families participated in the *CCT Oportunidades* program found that the subjective well-being of these adolescents was dependent on a positive self-concept (as defined by positive self-esteem and personal strength), as well as positive interactions with parents and their friends [22]. Because schools provide an opportunity for development and socialization, positive relationships fostered within the school environment may further incentivize school attendance. Thus, CCTs may strengthen children’s internal resources by incentivizing school attendance and preventative healthcare, which could enhance their academic abilities and physical health [21]. On the individual level, both UCTs and CCTs may help enhance health and school-related internal resources for youth.

### 3.2. Microsystem

CCT and UCT programs are directly influencing what Bronfenbrenner [54,55,56] would describe as the microsystem, which encompasses interactions between the children and their immediate environment, such as family and peers. Cash transfer programs directly target the family and thus contribute to changes in the family context [21]. One potential mechanism through which they do so is by relieving family stress, as described in the family stress model, which states that caregivers who are overwhelmed by the demands of poverty face understandable barriers to meeting the emotional and cognitive needs of their children [13].

In addition, cash transfers can enhance parenting self-efficacy, hope, and self-esteem by strengthening parents’ sense of their ability to support their children. The pilot UCT program in New York City provided families with a sense of adequacy, as they were able to provide for their children’s basic needs as they arose, such as buying a MetroCard (e.g., a card for public transportation via the subway) to get to a doctor’s appointment [39]. Similarly, a study that examined program participation in a UCT in Zambia found that such participation positively enhanced women’s attitudes toward their child’s future outcomes and well-being due to reduced perceived poverty [38]. Thus, it appears that one mechanism by which cash transfers can have a positive impact on parents/guardians is by providing them with a sense of hope and confidence in their ability to meet their children’s overall needs and to provide for their future.

Cash transfers may strengthen the quality and quantity of parental interactions by giving families access to income that can alleviate some of the poverty-related pressures that lead to parenting stress and poor mental health. This is evident, for instance, in research that indicates that enhanced parent well-being related to participation in UCTs has been linked to the alleviation of material hardships and the ability to meet basic family needs [35,39]. In addition, enhancements in parent mental health, such as reduced maternal depressive symptoms [29,35,36] and enhanced sense of parental subjective happiness and hope [38], may increase the quality of parent–child interactions and parental investment in their children. For instance, parents who feel less depressed and are in a happier and more hopeful state may be more likely to have greater emotional resources to look outside themselves to connect with family members such as children and spouses [15].

Peer interactions are also indirectly influenced in the child’s microsystem [54,55,56]. By incentivizing education, CCT programs effectively work towards changing norms around the importance of education, attendance in school, and school-related activities [57]. In doing so, they influence family norms and expectations that lead to changes in routines related to how much time children spend in school and on school-related activities. Having children participate with peers in the school setting is a way to support the development of social–emotional competencies in a structured environment with the support of teachers and school personnel. As a result, the CCTs foster children’s development and identity through interactions with peers in the school setting [58].

UCTs, while not directly targeting specific behaviors, have also been linked to positive peer relationships and decreases in risky behaviors. South Africa’s Child Support Grant (CSG), a UCT program sponsored by the South African government, focused on reducing poverty among youth by giving monthly payments to primary caregivers. One study conducted qualitative fieldwork that found that the CSG led to reductions in adolescent risky sexual activity and improved school–peer interactions [51]. By reducing adolescents’ perceptions of family needs and enhancing access to material needs, adolescent stress about social status and peer pressure was reduced.

### 3.3. Exosystem and Macrosystem

Finally, Bronfenbrenner’s [54,55,56] socioecological systems theory describes the exosystem and macrosytem. An application of the exosystem in the context of our discussion refers to interactions within the child’s greater community that do not directly involve the child, while the macrosystem reflects cultural attitudes and ideologies that might influence social policy.

While the exosystem and macrosystem are not direct targets of cash transfer programs, there are indirect mechanisms by which these programs function to enhance child and parent mental well-being. For example, in the exosystem, behavioral requirements for school attendance might help strengthen family ties to the school system and encourage increased interaction between caregivers and school administrators, which may be a protective factor within the exosystem [22]. As mentioned above, CCTs specifically function by changing family and peer norms around education and healthcare [21]. This can subsequently have an impact on communities and neighborhood-level norms in the macrosystem if CCT programs are being implemented on a larger population scale. For instance, if all or most families in a specific community are held to a similar set of standards around healthcare utilization and educational access, the community’s cultural norm can shift towards community investment in, and a value of, healthcare and education.

This culture shift may subsequently lead to positive community-level outcomes related to mental health, well-being, and mental healthcare utilization through greater utilization of services. By enhancing community-level interactions with schools and healthcare settings, CCT programs also have the potential to provide pathways to intervention for mental health, both through prevention and treatment, thus having a potential positive impact on the nature of the exosystem.

## 4. A Need to Address Stigma and Consider Structural Factors

While the aforementioned CCTs and UCTs indicate evidence of positive outcomes on the mental well-being of children and families, the results are not always consistent, with some studies reporting no associations between participation and externalizing or depressive symptoms [30,31,32], while others report significant reductions in depression and increased happiness [27,35,36,38,41,42]. These differing findings may be due to how studies measure the impact of mental well-being, with some studies not reporting on mental health effects because they were not a direct target of the cash transfer program. For instance, some studies measured the impact of mental illness by measuring maternal depression [35] or child behavioral aggression [27], while another study measured mental well-being factors such as maternal happiness and child hopefulness for the future [38].

Some research suggests that there can be a negative psychosocial impact of cash transfers related to social stigma and exclusion [59,60,61]. Although cash transfers are meant to help reduce the stigma associated with the immediate effects of poverty, such as by alleviating material hardships, being the recipient of income supplements may paradoxically lead to families feeling less independent or stigmatized by those around them. For instance, some qualitative research on UCT programs in the Middle East and Sub-Saharan Africa indicated that recipients reported decreased self-reliance and independence and feeling stigmatized [61].

Similarly, in an analysis of Peru’s CCT, Cookson (2018) found that stigmatization may result when caregivers internalize failure to comply with conditions [59]. Some CCT beneficiaries in Peru experienced shame when waiting in long lines to collect their cash transfer, as the collection site was in a busy and central location in the community [59]. However, there is scant empirical research demonstrating a connection between CCTs and shame or stigmatization [60].

Within the context of these contrasting results, while CCTs impose conditions to specifically target aspects related to intergenerational poverty, in doing so, it is also important to consider ways that they might be delivered such that they do not undermine dignity and self-efficacy [62]. Reduced parent self-efficacy can have negative implications for the family that may subsequently have an impact on the parent–child relationship. Thus, in considering broad implementation of these programs, it is important to consider possible negative implications on participants that may result from the additional income support. This awareness is important so that de-stigmatizing strategies can be implemented. For instance, one solution indicated by Rojas et al., 2020 was to distribute the income through a debit card to reduce visibility for program participants [39].

Cash transfer programs might also be seen as failing to address the structural factors that cause poverty and poverty-related disparities across different systems. Cash transfer programs at their core aim to reduce intergenerational poverty through targeting material needs, and in CCTs’ case behaviors and incentives, it seems important to address the limits of these programs given that poverty and the resulting negative impact on family well-being is also linked to larger structural factors [63]. Such structural factors may include income inequality, racism, and mental health-related stigma.

Considering structural factors is particularly important when focusing on poverty-related influences on child, parent, and family mental health. Cash transfer programs that target the individual and family do not necessarily address the structural factors mentioned previously. CCT programs may be especially flawed given that they target individual-level behaviors related to education and health, which will only be effective if educational institutions and healthcare are well resourced. For instance, what might be perceived as a demand problem (e.g., low enrollment rates in schools) may actually be a supply-side issue (e.g., lack of teachers or classrooms). In this case, the benefits of cash transfers for those in poorly resourced environments may be quite limited [64].

## 5. Overview and Summary

The current paper discusses the implications of cash transfer programs from a global perspective, regarding how they might promote positive mental health outcomes among families who face significant and multidimensional stressors. This paper presents the contextual reality that children across the world are disproportionally affected by extreme poverty. According to the 2022 World Bank’s extreme poverty line, an estimated 333 million children live on less than $2.15 per day [1]. This finding connects to mental health outcomes, as indicated by the substantial literature that document how childhood poverty may relate to poorer mental health outcomes among young people. These realities were further exacerbated by the COVID-19 pandemic. The family stress model [13] can be applied to this context, given its focus on how multiple stressors and economic pressure can disrupt family functioning.

In the context of severe stressors faced by children and families globally, our research question was to explore the finding that increasing financial resources for families through cash transfer programs can enhance emotional resources, thus promoting positive mental health outcomes among children and families. Through a consideration of conditional and unconditional cash transfer programs in Latin America, Africa, India, and North America, there appears to be less research regarding the impact of CCTs on child and adult mental health. Some studies offered promising results such as the aforementioned research that found children 4 to 6 years of age demonstrated decreases in aggression and oppositional behaviors after program participation [27]. Another study found that women from poor rural areas of Mexico who participated in the *Oportunidades* program reported decreased symptoms of depression [29]. A program in New York City highlighted potential implications for decreased risky behaviors in peer groups among adolescents [33]. Research on UCTs demonstrated a significant decrease in parental depression among parents in India and South Africa who participated in these programs [35,36].

While these findings underscore some evidence of positive outcomes that CCTs and UCTs may have on mental well-being among children and families, our theoretical application of Bronfenbrenner’s model also indicated that the research findings are not necessarily consistent. In other words, while some programs reported positive significant findings, other programs did not necessarily replicate findings. Such differing empirical outcomes may reflect factors such as researchers defining variables like mental health in different ways. Feelings of stigma and/or shame that families may experience in relation to the need of having to accept cash transfer program support may be another factor; however, more research is needed in this domain.

Bronfenbrenner’s [54,55,56] ecological model presents differing structural levels of engagement with the community. As mentioned, the model starts with the microsystem that addresses the experiences of children and their immediate environment and moves through to the macrosystem that reflects larger cultural ideologies, such as those that might influence social policy. The consideration of promising results amid mixed findings for cash transfer programs leads to the conclusion of the proposed research and policy agendas as described below.

## 6. A Call for Transnational Research to Explore Connections between Cash Transfer Programs and Mental Health Outcomes among Families 

### 6.1. Research Agenda

#### 6.1.1. Line of Inquiry

More research is needed that specifically looks at connections between cash transfer programs and mental health outcomes for children/adolescents and their parents/caregivers. If stigma related to enrollment in cash transfer programs is an issue, an interesting line of inquiry is to explore what might make enrollment in cash transfer programs more acceptable.

#### 6.1.2. Multi-Country Study

To explore this line of research in a more systematic, global way, it is suggested that countries partner develop a multi-site plan. In this way, countries create a transnational research team dedicated to the implementation of the research agenda in respective country sites. This allows for a multi-perspective approach when examining the potential mental health benefits of cash transfer programs for children and their families. The Swiss Commission for Research Partnerships [65] published a guidebook that presents recommendations for developing transnational research partnerships [66]. Their eleven principles include: “set the agenda together; interact with stakeholders; clarify responsibilities; account to beneficiaries; promote mutual learning; enhance capacities; share data and networks; disseminate results; pool profits and merits; apply results; [and] secure outcomes” [66] (p. 24).

In applying these guidelines to develop transnational partnerships, our understanding of the potential mental health benefits of cash transfer programs may be better operationalized and implemented in a more unifying way. Countries can then communicate with one another and share respective findings. In turn, the results of this body of work can be disseminated and connected with global organizations such as the World Health Organization and UNICEF.

### 6.2. Policy Agenda

It is important that a policy agenda accompany the proposed research structure given the findings revealed through our theoretical application. As mentioned, the target of cash transfer programs is at the level of the individual unit such as the child or family. While an increase in financial resources is important and thought to contribute to some of the well-being outcomes documented above, the individual nature of cash transfer programs means they do not address larger contextual issues such as racism, economic disparity, inequality, the wage gap, and many more. Perhaps inconsistent findings in the literature about the impact of cash transfer programs on mental health outcomes relate to the fact that cash transfer programs are implemented on a micro level, without consideration for the surrounding larger systemic context.

Thus, in addition to a proposed transnational research agenda, it is suggested that the research program be accompanied by a policy approach that moves cash transfer programs beyond micro-level considerations to a larger macro-level intervention. One recommendation, for instance, is that cash transfer programs be part of a larger governmental policy reform program. For instance, cash transfer programs can be one aspect of a social justice campaign designed to meet a societal need such as decreasing the wage gap, reducing unemployment, providing job training, or addressing systemic racism.

Hence, the aim of the policy agenda is to encourage governments to address macrosystem-level issues and at the same time, include the cash transfer program as one aspect of a systems plan. This might also decrease the potential stigma associated with accepting cash transfer programs while simultaneously elevating their potential impact if the larger social context is framed by policies and outreach to address macro-level issues.

Through a joint agenda that includes research to explore how cash transfer programs might further positive mental health outcomes for children and families—alongside a public policy effort that locates cash transfer programs within larger social policy reform—it is hoped that cash transfer programs will continue to be supported and studied so that we can learn more about the ways in which they promote positive mental health outcomes for children and families. The decision to adopt CCTs or UCTs is often influenced by political agendas, resource availability, and ideological perspectives within specific countries or regions [59]. Future research can unpack the political factors that influence the implementation of cash transfers in different regions throughout the world. While discussing the political determinants and implications of cash transfers is beyond the scope of this paper, the findings provide a call to action for governments who want to support positive mental health outcomes for children and families.

## Figures and Tables

**Figure 1 behavsci-14-00770-f001:**
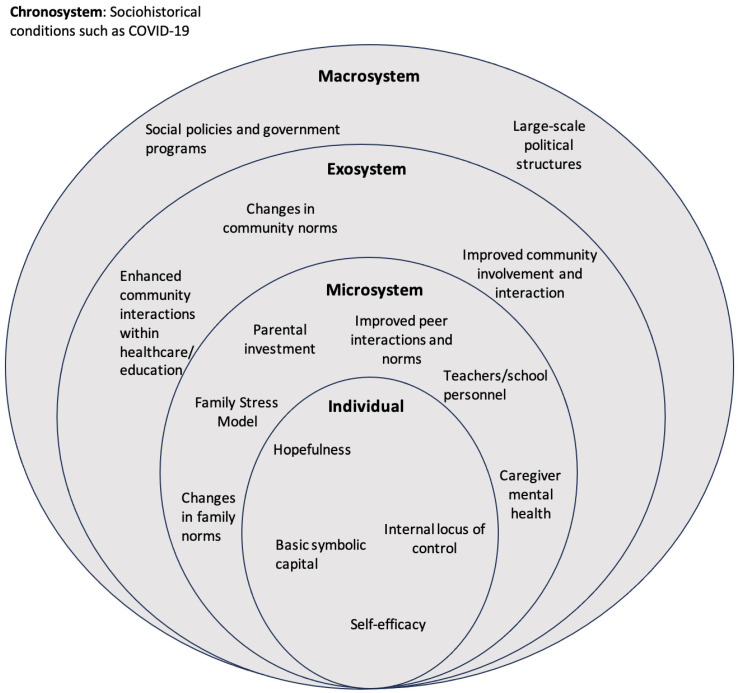
Implications of cash transfers on Bronfenbrenner’s ecological framework and child development and well-being.

## Data Availability

No new data were created or analyzed in this study. Data sharing is not applicable to this article.

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
