# Peer review of "Implications of Cash Transfer Programs for Mental Health Promotion among Families Facing Significant Stressors: Using Ecological Systems Theory to Explain Successes of Conditional and Unconditional Programs"

_behavsci, 2024, doi:10.3390/bs14090770_

Round 1
Reviewer 1 Report
Comments and Suggestions for Authors
The manuscript «Implications of Cash Transfer Programs for Mental Health Promotion in Low-Income Families: A Review of Conditional and Unconditional Programs” is a policy review of the potential impact of conditional and unconditional cash transfer programs. This is a very important topic that deserves a lot of attention. A potential reader might be left with some questions after reading, so I would like to ask the authors to try to address them in their manuscript.
· There might be a typo in line 233.
· When mentioning self-concept and self-esteem (e.g. in line 341) the authors should make it clear which of these concepts they are referring to and make it clear what effects on self-concept or self-esteem have previous studies found.
· The authors claim that due to poverty children might spend more time with peers therefore exposing them to more risky behaviors after they have claimed that extended family time might lead to more exposure to domestic violence. Clarification of the context in which these effects occur would help in understanding the processes.
· When discussing the expected effects of UCTs on child mental health the authors should try to compare them to the effects of CCTs.
· In general, the mechanisms behind the expected effects of UCTs should be compared with mechanisms behind the expected effects of CCTs, f.eks. on the individual level in the Bronfenbrenner’s socioecological systems theory.
· The authors focus on mental health and use Bronfenbrenner’s socioecological systems theory to explain the mechanisms by which these programs lead to positive outcomes on mental health, but when explaining these mechanisms, the authors focus on soc.-cognitive concepts and not on mental health concepts. The link to mental health should be clarified.
· The authors devote a lot of focus in the introduction to the effects of COVID-19 pandemic, and then these effects are watered down until they are not mentioned at all. This focus in the introduction seems too strong and not justified. The effect of COVID-19 on both mental health and poverty can be mentioned, but that does not seem related to the effects of CCTs and UCTs.
· The authors focus on the effects of poverty on mental health and only occasionally mention the effect of education on mental health. Both joint effects of poverty and education, and separate effects of poverty and separate effects of education should be discussed and can be linked to CCTs.
· When discussing the exosystem in use Bronfenbrenner’s socioecological systems theory no data or studies are mentioned. An overview of studies whose findings support expected effects of CCTs and UCTs on exosystem is needed.
· Throughout the manuscript the review of studies seems very selective and unsystematic. E.g. studies from sub-Saharan Africa, the Middle East and Peru are only mentioned around line 437 and it is not clear why have these studies not mentioned more earlier in the article. A more comprehensive overview of studies earlier in the manuscript would reduce this problem.
· The authors should justify the use of unsystematic over a systematic review. In addition, they should explain why the manuscript is described as a policy review and not as a literature review.
· The use of CCTs and UCTs seems to be determined by politics in different countries or states or regions. The political aspect of a choice of implementing or not CCTs and UCTs should be discussed in the manuscript.
Reviewer 2 Report
Comments and Suggestions for Authors
Thank you for giving me the opportunity to review the article. The title is very interesting, as well as the research question on whether the type of financial aid for vulnerable families, conditional or unconditional, has a different effect on the resilience and other variables of the target population.
The introduction is very well written and helps the reader to be aware of the importance in studying the different impact of both types of programs. However, I do not find how the most relevant programs have been selected in such extensive geographical areas (USA, Latin America, Africa and India); databases, keywords, description of the programs, criteria for selecting them, are missing.
The authors in a smart way present the variables contemplated in these programs organized in the Bronfenbrenner diagram; however, we do not have an adequate description of them that would allow the reader to understand this classification, and even replicate it.
The research would be greatly improved if the authors provided information on the selection of the programs, the methodology used, the variables that make them up, the time, etc. ; If this were provided, the results would be more substantiated, and consequently the conclusions.
I advise that the authors dedicate a section to the method, where they explain how they have carried out the search to find the programs, their contents, their criteria to include or exclude, finally the sample used, a more detailed description of these programs, including the variables, results, and based on this, to be able to replicate the research.
Additionally it is surprising the extensive geographical area used; it would be important to point out the reasons behind that led the authors to contemplate such a large area and at the same time so different socially, culturally and economically.
Round 2
Reviewer 1 Report
Comments and Suggestions for Authors
I would like to thank the authors for revising the manuscript. I can see that the manuscript has been updated, and I have some suggestions for improvement, that do not differentiate from my previous suggestions.
It should be clear from the title and abstract, and throughout the manuscript that the manuscript does not describe a study, that there are no methods, and that the review is not systematic. In the manuscript the authors should justify the use of unsystematic over a systematic review. Why was a systematic literature review avoided? In addition, they should explain why the manuscript is described as a policy review and in the next sentence a literature review. What kind of review is it needs to be clear and justified.
The point that the use of CCTs and UCTs seems to be determined by politics in different countries or states or regions has not been addressed, despite adding the last sentence to the manuscript. The political aspect of a choice of implementing or not CCTs and UCTs should be discussed in the manuscript.
Round 3
Reviewer 1 Report
Comments and Suggestions for Authors
Thank you for submitting the revised version of the manuscript. In the revised version you write that "using inclusion and exclusion criteria would be limiting for this paper". Could you specify in which way doing a systematic literature search is limiting. This needs to be clear to potential readers.
Author Response
Thank you for your helpful feedback. Below is the response to the comment “In the revised version you write that "using inclusion and exclusion criteria would be limiting for this paper". Could you specify in which way doing a systematic literature search is limiting. This needs to be clear to potential readers.”:
We added the following paragraphs starting on line 103 and highlighted them in green in the manuscript.
While a systematic review is useful for rigorous investigation and considered “the gold standard in evidence synthesis” [19] (p. 30), using absolute inclusion and exclusion criteria would be limiting for this paper when exploring the connections between financial re-sources, emotional resources, and mental health outcomes. The rationale for this approach corresponds with recent research that documents various limitations related to systematic reviews. This work highlights the importance of providing empirical evidence that identifies potential research flaws in systematic reviews, particularly given the fact that such reviews are often used to inform important decisions as described by Uttley and colleagues (2023).
Three of the specific problems with systematic reviews that were identified and defined by Uttley et al. (2023) apply to the current article’s approach to incorporating a non-systematic review that does not use inclusion and exclusion criteria. One systematic review flaw is the possibility of “overly stringent inclusion criteria affecting external validity” [19] (p. 36). This flaw corresponds with the current paper in that the use of a systematic review may have resulted in not including articles in the literature that reflected the impact and outcome of both CCT and UCT programs.
Another applicable potential flaw identified by Uttley and colleagues (2023) is that of “insufficient literature searches” (p. 36). Given the dearth of literature on CCT and UCT programs, a scarcity that is heightened when we consider articles that discuss mental health implications connected to these programs, employing a systematic review may have led to an insufficient number of review articles. This reality is compounded by the fact that the current paper’s non-systematic review addresses complex connections among variables such as CCT programs, UCT programs, economic resources, emotional resources, and mental health considerations.
Finally, a third related flaw that supported the rationale to use a non-systematic approach was the “failure to consider equity, different socioeconomic groups, or disadvantaged populations” [19] (p. 37). Because CCT and UCT programs are implemented to support challenges such as economic disparities, and given that such programs are implemented in diverse communities (see Table 1), the authors were concerned that a systematic review might not incorporate scholarship that reflects their diversity and global implementation. In other words, the limitation of a systematic review in this domain was the potential omission of articles that capture the impact and experiences of CCT and UCT programs among diverse global contexts.
We added the following reference:
Uttley, L., Quintana, D. S., Montgomery, P., Carroll, C., Page, M. J., Falzon, L., Sutton, A., & Moher, D. (2023). The problems with systematic reviews: A living systematic review. Journal of clinical epidemiology, 156, 30–41. https://doi.org/10.1016/j.jclinepi.2023.01.011
